# Evaluation of a New Extracorporeal CO_2_ Removal Device in an Experimental Setting

**DOI:** 10.3390/membranes11010008

**Published:** 2020-12-23

**Authors:** Matteo Di Nardo, Filippo Annoni, Fuhong Su, Mirko Belliato, Roberto Lorusso, Lars Mikael Broman, Maximilian Malfertheiner, Jacques Creteur, Fabio Silvio Taccone

**Affiliations:** 1Pediatric Intensive Care Unit, Bambino Gesù Children’s Hospital, IRCCS, 00165 Rome, Italy; 2Department of Intensive Care, Hôpital Erasme, Université Libre de Bruxelles (ULB), 1050 Brussels, Belgium; filippo.annoni@erasme.ulb.ac.be (F.A.); sufuhong@yahoo.com (F.S.); jacques.creteur@ulb.ac.be (J.C.); fabio.taccone@ulb.ac.be (F.S.T.); 3Anestesia e Rianimazione II Cardiopolmonare, Foundation IRCCS, Policlinico San Matteo, 27100 Pavia, Italy; m.belliato@gmail.com; 4Heart & Vascular Centre, Maastricht University Medical Centre, 6229 Maastricht, The Netherlands; roberto.lorussobs@gmail.com; 5ECMO Centre Karolinska, Karolinska University Hospital, 17164 Stockholm, Sweden; lars.broman@sll.se; 6Department of Physiology and Pharmacology, Karolinska Institutet, 17177 Stockholm, Sweden; 7Internal Medicine II, University of Regensburg, 93053 Regensburg, Germany; maxmalfertheiner@gmail.com

**Keywords:** extracorporeal CO_2_ removal, lung protective ventilation, mechanical ventilation, experimental model

## Abstract

Background: Ultra-protective lung ventilation in acute respiratory distress syndrome or early weaning and/or avoidance of mechanical ventilation in decompensated chronic obstructive pulmonary disease may be facilitated by the use of extracorporeal CO_2_ removal (ECCO_2_R). We tested the CO_2_ removal performance of a new ECCO_2_R (CO_2_RESET) device in an experimental animal model. Methods: Three healthy pigs were mechanically ventilated and connected to the CO_2_RESET device (surface area = 1.8 m^2^, EUROSETS S.r.l., Medolla, Italy). Respiratory settings were adjusted to induce respiratory acidosis with the adjunct of an external source of pure CO_2_ (target pre membrane lung venous PCO_2_ (P_pre_CO_2_): 80–120 mmHg). The amount of CO_2_ removed (VCO_2_, mL/min) by the membrane lung was assessed directly by the ECCO_2_R device. Results: Before the initiation of ECCO_2_R, the median P_pre_CO_2_ was 102.50 (95.30–118.20) mmHg. Using fixed incremental steps of the sweep gas flow and maintaining a fixed blood flow of 600 mL/min, VCO_2_ progressively increased from 0 mL/min (gas flow of 0 mL/min) to 170.00 (160.00–200.00) mL/min at a gas flow of 10 L/min. In particular, a high increase of VCO_2_ was observed increasing the gas flow from 0 to 2 L/min, then, VCO_2_ tended to progressively achieve a steady-state for higher gas flows. No animal or pump complications were observed. Conclusions: Medium-flow ECCO_2_R devices with a blood flow of 600 mL/min and a high surface membrane lung (1.8 m^2^) provided a high VCO_2_ using moderate sweep gas flows (i.e., >2 L/min) in an experimental swine models with healthy lungs.

## 1. Introduction

Significant advancements have been done to understand the feasibility and safety of extracorporeal CO_2_ removal (ECCO_2_R) in patients with hypoxemic and/or hypercapnic respiratory failure [1,2,3,4]. ECCO_2_R has been used either to reduce the main components of the mechanical power (i.e., respiratory rate, driving pressure, flow rate and/or positive end expiratory pressure), which may potentially cause ventilator-induced lung injury (VILI) in patients with severe acute respiratory distress syndrome (ARDS) [5], or, to avoid endotracheal intubation and invasive mechanical ventilation in patients failing non-invasive ventilation for acute exacerbations of chronic obstructive pulmonary disease (COPD) or of end-stage respiratory disease awaiting for a lung transplant [3,4,5,6,7]. For these purposes, several devices have been developed (i.e., pumpless arterio-venous or pump-driven veno-venous circuits), using a varying range of blood flow (i.e., from 200 to 1800 mL/min), sweep gas flow and different sizes (m^2^) of membrane lungs. Further, to increase CO_2_ removal with minimal blood flow, hybrid techniques have been developed. These techniques, using an acidified dialysate and a hemofilter, may increase the amount of H+ in the blood and consequently the CO_2_ content upstream of the membrane lung. So far, hybrid techniques (regional acidification) may increase the amount of CO_2_ removed; however, their complexity has limited their clinical use [8,9].

Recently, a multicenter pilot study conducted in patients with moderate ARDS showed that high-flow CO_2_ removal devices (i.e., blood flow around 800–1000 mL/min) had fewer hemorrhagic complications and hemolysis than low-flow (i.e., blood flow around 400 mL/min) devices, with a significantly better reduction of PaCO_2_ [2]. Animal data, instead, were controversial. Duscio et al. [10] reported very high CO_2_ removal (171 mL/min) using a low-flow device (400 mL/min), while Karagiannidis et al. [11,12] showed that only high blood flow rates (>900 mL/min) and adequate membrane lungs (surface area > 1 m^2^) can effectively correct severe respiratory acidosis. However, both studies converge on the point that the sweep gas flow can increase CO_2_ removal only when high blood flow rates are used.

With the present study conducted in healthy pigs, we aimed to describe the CO_2_ removal performance and operational characteristics of a new medium-flow ECCO_2_R device, which has been created specifically for CO_2_ removal using a fixed amount of blood flow rate, a membrane lung of 1.8 m^2^ and different sweep gas flow rates.

## 2. Methods

### 2.1. Extracorporeal CO_2_ Removal Technique

Medium-flow veno-venous ECCO_2_R was performed using the CO_2_RESET device (EUROSETS S.r.l., Medolla, MO, Italy). This device, driven by roller pumps, incorporates both a hemoperfusion membrane and a phosphorylcoline-coated polymethylpentene hollow fiber membrane lung (surface area = 1.8 m^2^), without an integrated heat exchanger. The membrane lung may be connected either to a sweep gas source of pure oxygen or to a mixture of air/oxygen to provide CO_2_ removal. The CO_2_RESET circuit is customized for single-use and can be connected to a wide range of cannulas. In our animal model, the hemoperfusion membrane was not incorporated and the device was used specifically for CO_2_ removal (Figure 1). The CO_2_RESET circuit is customized to receive only ¼ connectors and connects to either with dual lumen cannulas (13, 16, 19 French) or two single cannulas.

### 2.2. Animal Preparation

The Institutional Review Board of the Free University of Brussels (Belgium) approved the experimental protocol (number of Ethical Committee approval: 731N). On the day of the experiment, the animal (swine, *Sus Scrofa Domesticus*) was fasted for 12 h with free access to water. Anesthesia was initiated with a combined intramuscular injection of midazolam (1 mg/kg, Mylan, Auckland New Zeland and ketamine (100 mg/kg, Dechra, Lille, Belgium) administered in the neck and placed in supine position. The animal was monitored with a continuous electrocardiogram, and a peripheral vein (18-gauge) was inserted to provide a continuous infusion of sufentanil citrate (3 µg/kg, Janssen, Beerse, Belgium). A femoral 4.5 French (Fr) arterial catheter (Vygon, Ecouen, France) was inserted in the femoral artery and connected to a pressure transducer (True Wave, Edwards, CA, USA) for invasive arterial pressure monitoring and blood gas analysis (BGA). After a sequential intravenous injection of 1 mg atropine sulfate (Sterop, Anderlecht, Belgium), 3 µg/kg of sufentanil citrate and 1.2 mg/kg of rocuronium (Esmeron, MSD, Kenilworth, NJ, USA), an 8 mm endotracheal tube (Medtronic, Minneapolis, MN, USA) was placed and mechanical ventilation was started in controlled volume mode (Primus, Drägerwerk AG and Co, Frankfurt, Germany) with a tidal volume of 8 mL/kg, 5 cmH_2_O of positive end-expiratory pressure (PEEP), fraction of inspired oxygen (FiO_2_) of 1.0 and an inspiratory to expiratory time ratio of 1 to 2. A 1% mixture of inspired sevofluorane (Sevoflo, Abbott, Abbott Park, IL, USA) was started to achieve an expiratory percentage between 1.2 to 1.6%. Mechanical power of the respiratory system was calculated according to the validated formulas [13].

Ventilation parameters were subsequently adjusted to ensure an end tidal CO_2_ between 35 and 45 mmHg and SpO2 > 96%, using the minimally required FiO_2_. A continuous infusion of rocuronium (2–4 mg/kg/h) and sufentanil citrate (3.5 µg/kg/h) was maintained, and balanced crystalloids (PLASMA-LYTE, Baxter, Lessines, Belgium) were administered at a rate of 300–500 mL/h. A 14 Fr Foley catheter was surgically inserted to measure urine output thorough a midline incision in the lower abdomen, and the parietal layers were sutured separately. Under ultrasound guidance, a 5 Fr triple lumen central venous catheter (Arrow International, Reading, PA, USA) was placed in the right internal jugular vein and drug infusion was transferred to the distal line. For veno-venous ECCO_2_R, a 12 Fr multistage drainage cannula (REVAS, Free Life Medical GmbH, Aachen, Germany) was inserted in the left femoral vein and a 10 Fr return cannula (REVAS, Free Life Medical GmbH, Aachen, Germany) was inserted in the internal left jugular.

At this point, a continuous infusion of propofol (Propovet 2%, Abbott, NJ, USA) at the dose of 2–3 mg/kg was started, allowing for a progressive decrease until complete arrest of the anesthetic gas flow. The respiratory system was then opened with the removal of the soda lime absorber (Drägersorb Free, Drägerwerk AG and Co, Frankfurt, Germany). At the end of the preparation, the animal was proned and stabilized on the surgical table. For anticoagulation, heparin infusion was adjusted to achieve an activate clotting time (ACT) between 180 and 220 s, monitored via a dedicated point of care device (i-STAT, Kaolin ACT, Abbott Park, IL, USA). The estimated CO_2_ production in pigs at rest is about 200–250 mL/min [14], which is comparable to an adult human.

### 2.3. Study Design and Experiment Procedure

Considering the high reproducibility of the study and the request to avoid unnecessary use of animals from the Institutional Review Board for Animal Care (IRBAC), three healthy pigs were included in this study. After oral intubation and induction of respiratory acidosis by reducing mechanical ventilation settings, the pig was connected to the veno-venous ECCO_2_R device (CO_2_RESET, EUROSETS S.r.l., Medolla, MO, Italy).

Respiratory acidosis was induced with a 50% reduction of both the baseline tidal volume (from 8 to 4 mL/kg) and the respiratory frequency, to achieve an end tidal CO_2_ (etCO_2_) between 50 and 60 mmHg (Figure 2). To ensure a pre-membrane lung CO_2_ (P_pre_CO_2_) between 80–120 mmHg, an external supplementation of 1 L/min CO_2_ was provided to the inspiratory side of the ventilator circuit [15,16]. FiO_2_ was adjusted to maintain a SpO_2_ > 96%. A pre-membrane lung blood gas analysis (BGA) to control the achievement of this target and an arterial BGA from the animal were undertaken before starting each step of experiment. At this time, the veno-venous ECCO_2_R device was connected with a fixed blood flow rate of 600 mL/min and a sweep gas flow of 0 L/min (i.e., no CO_2_ removal capacity). All the experiments started with P_pre_CO_2_ between 80 and 120 mmHg. The experiment included six steps from 0 to 1, 2, 3, 5 and 10 L/min of sweep gas flow, respectively. Blood flow was fixed at 600 mL/min during all the steps. Each step of sweep gas flow was maintained for 30 min, and at the end, a post-membrane lung BGA and an arterial BGA from the animal were sampled and CO_2_ elimination (VCO_2_) was collected. At the end of each step, the sweep gas flows were brought to 0 L/min until the animal reached a P_pre_CO_2_ between 80 and 120 mmHg. Body temperature was maintained stable at 37 °C during the study using a warming blanket (Bair Hugger 3M, Zwijndrecht, Belgium). VCO_2_ was calculated directly from the device (multiplying the specific sweep gas flow for the partial pressure of the CO_2_ exhaled from the membrane lung) and provided in BTPS (body temperature, pressure, water vapor saturated). Operational characteristics of the ECCO_2_R device, including access, return and pressure drop across the membrane lung, were recorded at each step. Experiments were performed in each pig in a standardized fashion. At the end of the experiment, the animal was sacrificed by injection of 80 mEq of KCl under deep sedation.

### 2.4. Statistical Analysis

Data are expressed as median and inter-quantile ranges. Descriptive statistical analysis for non-parametric data was performed with Wilcoxon and Friedman test with Dunn’s multiple comparisons using GraphPad Prism 8 (San Diego, CA, USA). A *p* value < 0.05 was defined as statistically significant.

## 3. Results

Total duration of experiment was 6.30 (6.00–7.00) hours. Baseline characteristics of the studied animals (Table 1) were weight 54.00 (50.00–56.00) kg; static compliance of the respiratory system 34.00 (28.00–36.00) cmH_2_O/mL; driving pressure 13.00 (11.00–15.00) cmH_2_O; respiratory rate 18.00 (17.00–20.00) breaths/minute; minute ventilation 7.74 (6.80–9.00) L/min; PEEP 5.00 (5.00–5.00) cmH_2_O; mechanical power of the respiratory system 12.18 (10.00–13.25) Joule/minute; PaO_2_/FiO_2_ 480.00 (460.00–512.00); pH 7.44 (7.35–7.48) and PaCO_2_ 43.00 (39.00–44.00) mmHg. After the reduction of the ventilator settings and before the initiation of ECCO_2_R (gas flow 0 L/min), we observed (Table 1) a decrease of static compliance (28.00 (20.00–30.00) cmH_2_O/mL), driving pressure (8.00 (7.00–10.00) cmH_2_O), respiratory rate (9.00 (8.00–10.00) breaths/minute), minute ventilation (1.80 (1.72–2.24) L/min); mechanical power (1.91 (1.66–2.15) Joule/minute), pH (7.19 (7.16–7.25)) and PaO_2_/FiO_2_ (400.00 (380.00–450.00)); (*p* = 0.25 for all vs baseline); an increase of FiO_2_ (0.35 (0.25–0.40)); and *p* = 0.25 vs. baseline.

At the beginning of the experiments, P_pre_CO_2_ was 102.50 (95.30–118.20) mmHg and animal PaCO_2_ was 99.50 (88.10–105.00) mmHg. VCO_2_ progressively increased with an hyperbolic shape from 0 mL/min (gas flow: 0 L/min) to 90.00 (88.00–93.00) mL/min (gas flow: 1 L/min), 140.00 (138.00–170.00) mL/min (gas flow: 2 L/min), 153.00 (141.00–186.00) mL/min (gas flow: 3 L/min), 150.00 (145.00–190.00) mL/min (gas flow: 5 L/min) and 170 (160.00–200.00) mL/min (gas flow: 10 L/min); *p* < 0.001, (Figure 3 and Appendix A). VCO_2_ did not significantly increase during each step of increase of the sweep gas flow (*p* > 0.99, respectively, with Dunn’s multiple comparisons test). At the end of each step, animal PaCO_2_ was 71.80 (67.90–84.20) with 1 L/min gas flow, 69.50 (58.80–80.00) with 2 L/min gas flow, 68.50 (56.80–75.60) with 3 L/min, 66.70 (52.60–74.80) with 5 L/min and 66.00 (56.50–69.10) with 10 L/min (*p* = 0.25 respectively vs. baseline).

P_pre_CO_2_ and P_post_CO_2_ at each step of the experiment are shown in Figure 4. Decrease of P_post_CO_2_ at each step of the experiment was not significant (*p* > 0.99, *p* = 0.25, *p* = 0.25, *p* = 0.25, *p* = 0.25, *p* = 0.25, respectively). Operational characteristics of the ECCO_2_R device are reported in Table 2. During the experiments, the animals did not report any problem of bleeding or clotting. Pressure drops across membrane lung were 32.00 (30.00–34.00) mmHg. Median platelet count, fibrinogen and D-dimer levels remained stable during all the experiments (213.00 (185.00–230.00) cells/mm^3^, 180.00 (150.00–210.00) mg/dL and 215.00 (180.00–350.00) ng/mL, respectively). Levels of free plasma hemoglobin were undetectable (within normal < 50 mg/dL).

## 4. Discussion

The main finding of the present porcine study is that the use of a high-surface membrane lung may substantially impact CO_2_ elimination with a stepwise increase of the sweep gas flow rate despite using a fixed relatively blood flow rate (i.e., 600 mL/min). Pressure drops using these settings were within normal ranges. Although the experiment was conducted in a small cohort of healthy animals and with a limited time, no complications or technical issues were reported during the short observational period.

From a technical point of view, CO_2_ elimination may be manipulated at the bedside by modifying the blood flow rate, the sweep gas flow or the size of the membrane lung, or by acidifying the blood before the membrane lung (increase of P_pre_CO_2_). Animal data conflict with one another. Karagiannidis et al. [11,12] demonstrated, using a porcine model, that a blood flow rate > 1 L/min with a membrane lung with a surface area > 0.8 m^2^ may remove the 50% of total CO_2_ production and correct severe respiratory acidosis. Contrarily, Duscio et al. [10] reported a very high VCO_2_ (171 mL/min) with a low blood flow (400 mL/min) and a high surface membrane lung (1.8 m^2^).

Our results are in between the previous findings of Karagiannidis et al. [11,12] and Duscio et al. [10]. In our porcine model, using a fixed blood flow rate of 600 mL/min and a high surface membrane lung (1.8 m^2^), we reported a CO_2_ elimination of 170.00 (160.00–200.00) mL/min using 10 L/min of sweep gas starting with a P_pre_CO_2_ around 102.50 (95.30–118.20). Furthermore, CO_2_ elimination progressively increased when increasing the gas flow from 0 to 2 L/min and reached a plateau at sweep gas flows higher than 3–4 L/min, in line with other previous studies [11,16].

Using this configuration, our VCO_2_ was similar to the ones reported by Karagiannidis et al. [11], which used the same range of P_pre_CO_2_, higher blood flow rates (1 L/min) and smaller membrane lungs (0.8–1.3 m^2^), similar to the ones of Duscio et al. [10], which used a lower blood flood (400 mL/min) with a very high membrane surface area (1.8 m^2^), and higher than the ones reported by Hospach et al. [15], which used a blood flow of 600 mL/min with the PrismaLung (Baxter, Lessines, Belgium) and the A.L.ONE (EUROSETS S.r.l., Medolla, MO, Italy) membrane lungs (0.80 and 1.35 m^2^, respectively).

The use of low/medium-flow ECCO_2_R devices with this setting may have some advantages compared to high-flow ECCO_2_R devices. First, low/medium-flow ECCO_2_R are generally driven by roller pumps and have been developed specifically to remove CO_2_ [15]. High-flow devices are generally centrifugal pumps designed for higher blood flows and used for extracorporeal membrane oxygenation (ECMO). When these ECMO pumps are adapted to work at a lower blood flow rates and with smaller membrane lungs (neonatal or pediatric), these “adjustments” are not free of risks and may induce platelets activation and/or destruction as well as hemolysis, due to the reduction of the hydraulic efficiency and the increase of both pump recirculation rate and shear stress [17]. Second, circuit priming is fast and similar to the ones used for renal replacement therapy (RRT); furthermore, it does not require dedicated specialists as for ECMO (i.e., perfusionists). Third, low/medium-flow ECCO_2_R devices can integrate other organ support techniques (i.e., RRT) to promptly manage the patients with both acute respiratory failure and acute kidney injury/fluid overload. Together with these potential advantages, some challenges exist for low/medium-flow ECCO_2_R devices. First, to provide adequate level of VCO_2_ using a low blood flow, they require a large membrane surface area [10]. The interaction between a large membrane surface area and a low blood flow may induce the development of areas of blood stagnation, increasing the risk of thrombotic complications (“circuit” diffuse intravascular coagulation) and secondary hemolysis. Second, low/medium-flow ECCO_2_R devices require an external heating system to maintain body temperature within normal ranges.

This study presents some limitations. First, due to the ethical concerns raised by our IRBAC, few animals have been included; thus, our data cannot be directly transferred to humans with acute respiratory distress syndrome (ARDS) or decompensated COPD. Further, the blood and the interstitial fluid account only for less than 20% of the total human CO_2_ stores that can be mobilized within 48 h [18]. Thus, adding CO_2_ with an external source for a limited amount of time is not the most accurate strategy to increase CO_2_ storages [18]. However, we adopted this approach, previously used by other authors [15,16] to rapidly increase P_pre_CO_2_, avoiding the reduction of tidal volume (<4 mL/kg) and respiratory rate to unsafe levels that could have required an adjustment of the ventilator settings.

Second, only one fixed blood flow (600 mL/min) was used in our experiments, although the CO_2_ elimination is known to increase with the increase of blood flow; the operating range of CO_2_RESET is wider with a maximum of 800 mL/min. However, our purpose was to maintain a blood flow that was comparable with the ones described in other studies [15,17]. Of note, we preferred not to use lower blood flows (i.e., 400 mL/min) [10], to avoid the use of higher ACT (300 sec) for anticoagulation [1].

Third, our study has been designed to maintain a wide range of high P_pre_CO_2_ and consequently provides high VCO_2_. This wide range, even though not common when using ECCO_2_R to prevent VILI in ARDS patients, has also been chosen to test this new ECCO_2_R device in extreme clinical situations such as near fatal asthma and acute exacerbation of COPD or end-stage lung diseases awaiting a lung transplant [7].

Fourth, even though we did not report any mechanical complications (bleeding, clotting, air embolism, circuit failure or hemolysis) during the three experiments, we cannot draw any clinical conclusions since the duration of our experiments was limited.

## 5. Conclusions

Medium-flow ECCO_2_R devices with a blood flow of 600 mL/min and a high surface membrane lung (1.8 m^2^) provided high VCO_2_ using moderate sweep gas flows (i.e., >2 L/min) in an experimental swine model with healthy lungs. Future clinical data from these devices would provide further information on this approach in a human setting.

## Figures and Tables

**Figure 1 membranes-11-00008-f001:**
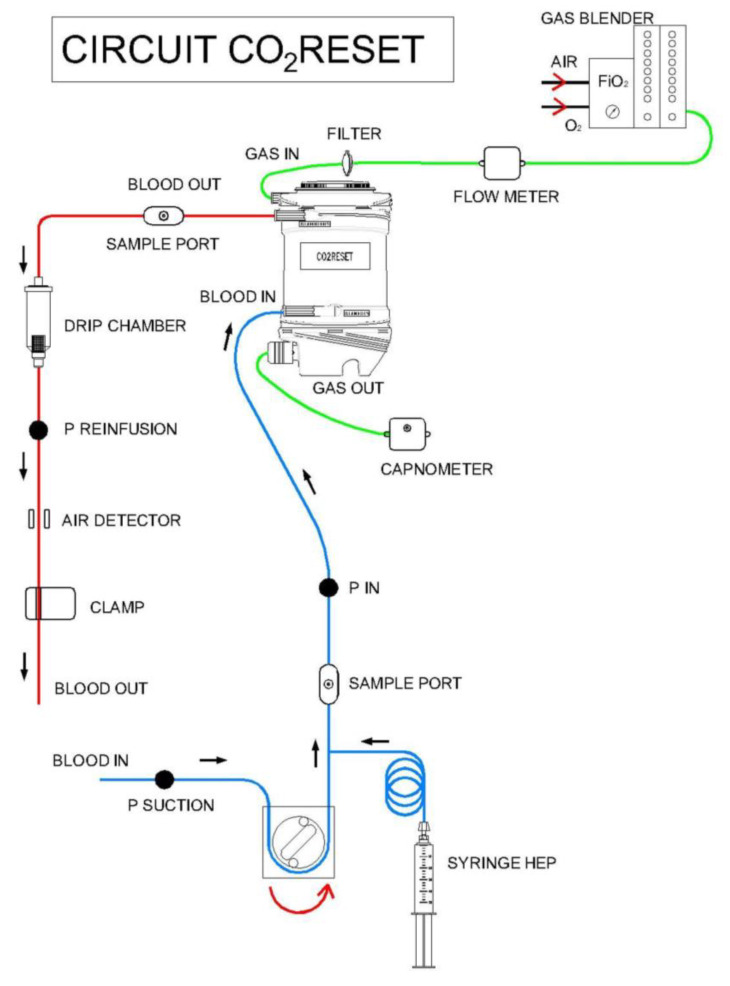
Schematic representation of the CO_2_RESET device. P = pressure.

**Figure 2 membranes-11-00008-f002:**
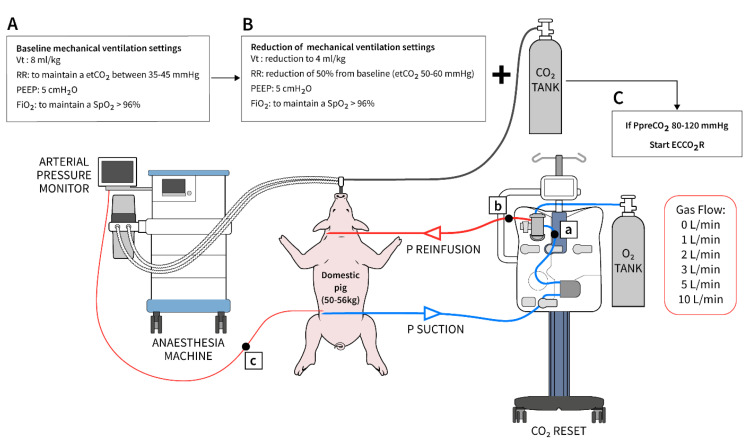
Diagram showing the experiment steps (**A**→**C**). a: sampling site pre-membrane lung; b: sampling site post-membrane lung; and c: arterial sampling site.

**Figure 3 membranes-11-00008-f003:**
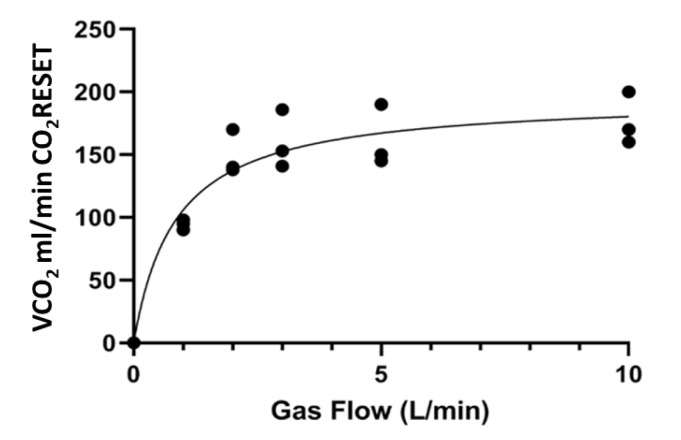
The relationship between sweep gas flow rate and the median CO_2_ elimination (VCO_2_) measured by the CO_2_RESET device with a fixed blood flow rate of 600 mL/min.

**Figure 4 membranes-11-00008-f004:**
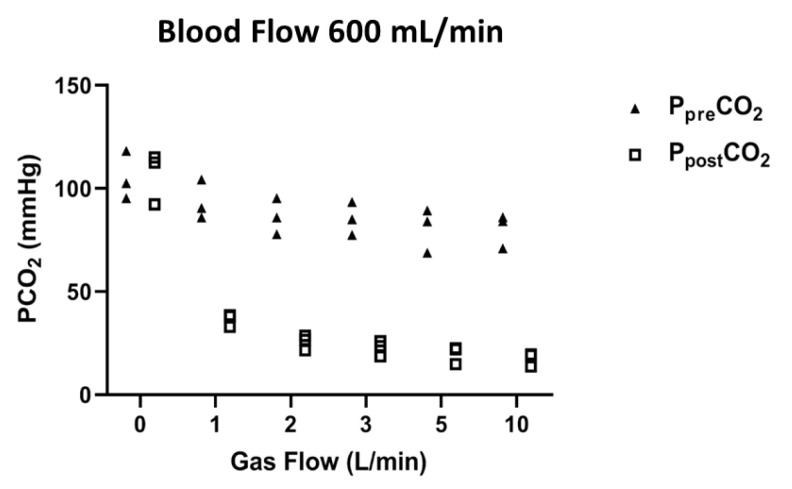
Median pre and post-membrane PCO_2_ under different sweep gas flow conditions (0, 1, 2, 3, 5 and 10 L/min) and a fixed blood flow rate of 600 mL/min.

**Table 1 membranes-11-00008-t001:** Main physiologic variables at baseline (Time 1) and at the beginning of the experiment (Time 2).

	Time 1	Time 2
Respiratory rate (breaths/min)	18 (17–20)	9 (8–10)
Tidal volume-pig (mL)	430.00 (400.00–450.00)	216.00 (200.00–224.00)
Minute ventilation (L/min)	7.74 (6.80–9.00)	1.80 (1.72–2.24)
Positive end-expiratory pressure (cmH_2_O)	5.00 (5.00–5.00)	5.00 (5.00–5.00)
Compliance respiratory system (cmH_2_O)	34.00 (28.00–36.00)	28.00 (20.00–30.00)
Respiratory system mechanical power (J/min)	12.18 (10.00–13.25)	1.91 (1.66–2.15)
Heart rate (beats/min)	73 (65–85)	77 (70–81)
Central venous pressure (mmHg)	8.00 (6.00–10.00)	9.00 (6.00–10.00)
Mean systemic arterial pressure (mmHg)	92 (88–100)	93 (87–99)
Arterial lactates (mmol/L)	1.20 (0.90–1.60)	1.35 (0.88–1.55)

**Table 2 membranes-11-00008-t002:** Operational characteristics of CO_2_RESET in the three treated pigs.

Characteristics	Value
Access pressure (mmHg)	−7.00 (−10.50–−3.50)
Pre-membrane pressure (mmHg)	40.00 (46.75–69.50)
Post-membrane pressure (mmHg)	17.00 (15.75–32.00)
Δ pressure (mmHg)	32.00 (30.00–34.00)
Activate Clotting Time	187.00 (184.00–191.00)

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
