# Peer review of "Evaluation of a New Extracorporeal CO2 Removal Device in an Experimental Setting"

_membranes, 2020, doi:10.3390/membranes11010008_

Round 1
Reviewer 1 Report
The authors report an experimental work on medium-flow ECCO2R with a rather new machine. While I understand the reason to perform this kind of work, I am not sure that the setup is actually adequately refined to provide insightful new information. The aim of an animal experiment should be to perform procedures and/or measurements that on human patients may not be strictly safe/justified. For this reason, I would have expected to see much more data, also given the really limited number of animals used. I am perfectly aware of the difficulties that we everyday encounter to obtain even a ridiculously low number of animals to perform experiment and, as such, I am not bothered by that. But given the rather low effort that 3 animals for few hours each carry, I paranoic attention to the details is required.
Here my comments:
Introduction:
- I would add two important points here: I would mention all the research line on regional acidification (electrodialysis and similar) and a recent paper from Duscio et al CCM, where an impressively high VCO2 was obtained from 400 ml/min. This is worth discussing also later in the discussion.
- "Mimicking CO2 retention": I would be very careful about this point. This is indeed the major intervention of your study, but it is highly controversial. Please read Giosa et al AJRCCM of a couple of months ago on CO2 retention and mobilization. It appears to be an extremely complicated topic and breathing CO2-enriched air for few hours is far from being close to what happens in pathological conditions.
Methods:
- You targeted the PETCO2 between 35 and 45. It is an enormous gap in terms of alveolar ventilation. This kind of experiments must be done at fixed PETCO2 or, even better PaCO2. Between 35 and 45 is a really large interval and definitely another major problem of your setup.
- This setup requires a double cannulation: I think it is worth discussing later pros and cons of a setup with lower flow and a single dual lumen cannula
- Natural lung VCO2 was not measured: I think that in this setup this measurement is key. It is key to understand what percentage of the VCO2 you are removing, how much the ventilation can be lowered, how much is the total CO2 production, the natural lung metabolic quotient (important to set the FiO2) and so on.
- I think table/figure with the experimental procedure/steps is requred.
- Maintain a "PpreCO2" between 80 and 120. Again, these ranges are far too wide.
- The membrane lung VCO2 obtained in this way is STPD or BTPS? It is important to specify. It is not trivial because blood contents are BTPS (normally) and gas flow in something else (STPD, ATPS maybe even other). If you make the calculations you will realize that there is quite a relevant difference
Results:
- The compliance of the animals is very low and it does not match my experience: were the animals prone or supine?
- The fact that VCO2 was not statistically different at higher gas flows while Post PCO2 decreased is probably due to the small sample size. From what we know till 8-10 l/min VCO2 should increase.
- I find a bit strange not to find a Table with all the ventilation, hemodynamic, gas exchange, lung mechanics, ECCO2r parameters at each step. Maybe I would present each animal separately. Without that it is difficult to reason on the numbers. For example: how was the minute ventilation at the beginning and at the end?
- Page 6, line 176: relatively?
Discussion:
- Medium flow devices do not require active heat exchange: First, I could not find the temperature of the animals at different steps. Second, did you put an heating blanket on them? In my experience, even much lower flows lead to a decrease in body temperature.
- Page 7 line 223: How can you say "with a normal CO2 production" if you did not measure it?
Overall my main criticisms are towards a relatively blurred and not clear/rigorous study protocol and, most importantly, a general lack of important measurements that are key to understand the paper.
Author Response
Reviewer 1
The authors report an experimental work on medium-flow ECCO2R with a rather new machine. While I understand the reason to perform this kind of work, I am not sure that the setup is actually adequately refined to provide insightful new information. The aim of an animal experiment should be to perform procedures and/or measurements that on human patients may not be strictly safe/justified. For this reason, I would have expected to see much more data, also given the really limited number of animals used. I am perfectly aware of the difficulties that we everyday encounter to obtain even a ridiculously low number of animals to perform experiment and, as such, I am not bothered by that. But given the rather low effort that 3 animals for few hours each carry, I paranoic attention to the details is required.
We thank the reviewer for the precious suggestions that were used to improve the quality of our paper. We fully agree with the reviewer that more physiologic details would have improve the quality and understanding of our results, thus we included in our revised manuscript a table showing physiologic parameters of lung mechanics, hemodynamic, and metabolism at baseline and at the beginning of the experiment.
We also included in the supplement the VCO2 of each pig being the PpreCO2 so high and with a wide range (PpreCO2 between 80-120 mmHg). This important aspect has been added in the limit of the study
For sake of clarity, we would like to underline that our main goal was to provide technical details of this new ECCO2R device using an animal model to simulate hypercapnia. To do this, we followed previous experiences published in literature to render comparable our model. This is why we adopted the use of an external supply of CO2 (Hospasch et al ICM experimental 2020, Barret N et al. Perfusion 2019) to increase blood PpreCO2 or we used a wide range of PpreCO2 (Strassman et al Intensive Care Experimental 2019).
We acknowledged these limits and we included them in the limit of the study of the revised manuscript.
Here my comments:
Introduction:
- I would add two important points here: I would mention all the research line on regional acidification (electrodialysis and similar) and a recent paper from Duscio et al CCM, where an impressively high VCO2 was obtained from 400 ml/min. This is worth discussing also later in the discussion.
Thanks for this comment. The introduction has been improved as well as the discussion, including the important of Duscio E. et al. and of Giosa L. et al. Regional acidification has been also included to provide a better understanding of the topic
- "Mimicking CO2 retention": I would be very careful about this point. This is indeed the major intervention of your study, but it is highly controversial. Please read Giosa et al AJRCCM of a couple of months ago on CO2 retention and mobilization. It appears to be an extremely complicated topic and breathing CO2-enriched air for few hours is far from being close to what happens in pathological conditions.
We fully agree with the Reviewer about the complexity of the CO2 storage in humans and that CO2 storages of pigs are not the same of humans (even though very close). These concerns which have been very elegantly discussed by Giosa et al have been included in the limits of our study. We also acknowledge that real life physiology is much more complex than just adding some CO2 in inspiratory branch of the ventilator. However, this was the approach chosen by our group for this experiment because we were more familiar to it (see below). Further, External supplementation of CO2 was also used by Hospach et al (ICM experimentals 2020) in their experiments to test different types of membrane lungs for CO2 removal.
Thus, to increase CO2 in our pigs we did not used only a 50% reduction of minute ventilation (4 ml/kg ) and halved respiratory rate, but we also added CO2 in the inspiratory branch of the ventilator. This was more feasible in our hands and well tolerated by the animals (no need to increase PEEP to counterbalance the development of atelectasis or increase FiO2 because of hypoxia). Further, the addiction of CO2 is commonly used in our clinical experience with neonates and infants when weaning from VA-ECMO. Sometimes during VA ECMO, CO2 removal is too high and cannot be controlled just reducing at minimum the sweep gas flow and EC blood flow. In all these cases to avoid excessive CO2 removal and impairment of cerebral blood flow, we used to add a fixed amount of CO2 with an external supply.
Methods:
- You targeted the PETCO2 between 35 and 45. It is an enormous gap in terms of alveolar ventilation. This kind of experiments must be done at fixed PETCO2 or, even better PaCO2. Between 35 and 45 is a really large interval and definitely another major problem of your setup.
We fully agree with the Reviewer. The range used was too wide and can be misleading when interpreting the median VCO2 presented in our first draft. For these reason, we have provided in the supplement the VCO2 of each pig and we have added this major point in the limit of the study.
- This setup requires a double cannulation: I think it is worth discussing later pros and cons of a setup with lower flow and a single dual lumen cannula
Thanks for this important comment. For sake of clarity, in our experiments, we used two sites cannulation just because the dual lumen cannulas are actually more expensive. This new device, however, can be also used with 13F, 16 F and 19F dual lumen cannulas. This point has been better addressed in the methods to avoid misunderstandings.
- Natural lung VCO2 was not measured: I think that in this setup this measurement is key. It is key to understand what percentage of the VCO2 you are removing, how much the ventilation can be lowered, how much is the total CO2 production, the natural lung metabolic quotient (important to set the FiO2) and so on.
Thanks for this comment. We fully agree with the Reviewer that without VCO2 of the native lung we cannot evaluate the total VCO2 production of our pigs. However, this was not the aim of our study.
Differently from the study of Duscio et al, which was more focalized on how much “ventilation” could be reduced and thus provides an accurate partitioning between native and membrane lung, our main objective was to study how much CO2 could be removed by this new device. In practical, we were interested in the efficiency of the new ECCO2R device.
For this reason, we preferred to use a cylinder to increase CO2 rather than further reduce the minute ventilation to limits that are far away from the clinical practice. In our clinical experience, reducing minute ventilation to very low limits was often associated to an important increase of shunt in injured lungs which was not often easily controlled by an increase of PEEP and FiO2.
- I think table/figure with the experimental procedure/steps is requred. This figure has been provided in the revised manuscript
- Maintain a "PpreCO2" between 80 and 120. Again, these ranges are far too wide. See above comments
- The membrane lung VCO2 obtained in this way is STPD or BTPS? It is important to specify. It is not trivial because blood contents are BTPS (normally) and gas flow in something else (STPD, ATPS maybe even other). If you make the calculations you will realize that there is quite a relevant difference.
VCO2 is obtained directly from the device. According to the manufacturer VCO2 is provided in BPTS.
We acknowledge that this is an important aspect often forgotten in research that may made the difference if not well addressed in advance.
Results:
- The compliance of the animals is very low and it does not match my experience: were the animals prone or supine? Thanks for this comment. Yes, this is true. Compared with other published data, our animals presented lower compliance. We do not have any explanation for that. The animals in these experiments were kept prone just for practical reasons. This aspect of body position has been well addressed in the methods of the study
- The fact that VCO2 was not statistically different at higher gas flows while Post PCO2 decreased is probably due to the small sample size. From what we know till 8-10 l/min VCO2 should increase. Thanks for this comment. Yes, this is probably due to the limited sample size. However, the main result from our study is that VCO2 tent to increase till 2L/min and then tend to slow. A similar result was also shown by Barret N et al. (Perfusion 2019)
- I find a bit strange not to find a Table with all the ventilation, hemodynamic, gas exchange, lung mechanics, ECCO2r parameters at each step. Maybe I would present each animal separately. Without that it is difficult to reason on the numbers. For example: how was the minute ventilation at the beginning and at the end?
Thanks, for this comment. As we pointed out the main aim of the study was to study the performance of this new ECCO2R device (VCO2 and operational characteristics). Thus, we focalized our attention on that. However, to help readers to reason on our data, we presented in the supplement a table showing the characteristics of the animals at baseline and at the beginning of the experiment.
- Page 6, line 176: relatively? This word was removed
Discussion:
- Medium flow devices do not require active heat exchange: First, I could not find the temperature of the animals at different steps. Second, did you put an heating blanket on them? In my experience, even much lower flows lead to a decrease in body temperature. Thanks for highlighting this important aspect. Yes, all animals were kept warm with a heating blanket. This aspect has been included in the methods
- Page 7 line 223: How can you say "with a normal CO2 production" if you did not measure it?
Thanks for this comment. This is right. Having not measured VCO2 of the native lung, we cannot say this.
Thus, this sentence has been removed.
Overall my main criticisms are towards a relatively blurred and not clear/rigorous study protocol and, most importantly, a general lack of important measurements that are key to understand the paper.

Reviewer 2 Report
In this study the authors investigated in 3 pigs the CO2 removal capacity of a recently developed external veno-venous CO2 removal device using a fixed blood flow and varying gas sweep flows. They found a rapid rise in CO2 removal up to 2 litres, and a much slower rise for higher gas sweep flows.
I have no fundamental concerns with this study. Please check the text on typo’s and remove decimals that suggest a too high precision (for example Table 1).
Author Response
Reviewer 2
In this study the authors investigated in 3 pigs the CO2 removal capacity of a recently developed external veno-venous CO2 removal device using a fixed blood flow and varying gas sweep flows. They found a rapid rise in CO2 removal up to 2 litres, and a much slower rise for higher gas sweep flows.
I have no fundamental concerns with this study. Please check the text on typo’s and remove decimals that suggest a too high precision (for example Table 1).
Thanks for reviewing our manuscript. Typos have been fixed accordingly.

Reviewer 3 Report
In this experimental study, De Nardo and colleagues evaluate the CO2 removal capacity of a novel device in a swine model. The principal interest of the study lies in the evaluation of this new device. It does not really add new generalizable information to the existing body of knowledge on extracorporeal CO2 removal.
MAJOR COMMENTS
(1) Comparisons with other ECCO2R membrane performances in the literature should be more thorough, detailed and explicit in the discussion. For instance, authors demonstrate a membrane lung VCO2 of 150 ml/min with a moderate blood flow of 600 mL/min. In a similar experiment, Duscio E and colleagues were able to demonstrate a similar VCO2 with a membrane of the same surface area with an even lower ECBF (400 ml/min). (Duscio E et al. Extracorporeal CO2 Removal: The Minimally Invasive Approach, Theory, and Practice, Crit Care Med. 2019 Jan;47(1):33-40.doi: 10.1097/CCM.0000000000003430)
(2) A limitation of the study is that its design maximizes the membrane VCO2 by creating a very high inflow PaCO2. For a given ECBF, membrane surface area, and sweep gas flow, the VCO2 of a membrane lung will be highly dependent on the inflow PaCO2. (the higher the inflow PaCO2, the higher the VCO2). (Duscio et al CCM 2019) In their experiment, De Nardo and colleagues maintained inflow PaCO2 high (about 100 mm Hg) by the addition of CO2 to the inspired gas. This will result in an overestimation of VCO2 to be expected in a real life scenario, where clinicians would rarely aim to maintain a pH of 7.19 and a PaCO2 at 100 mmHg. This should be added to the limitations section of the discussion.
MINOR COMMENTS
(3) Page 5, line 156: “Even though not significant, […]”. What test was used? Please provide p-value in the text.
(4) Page 5, line 158: “values remained between 85.95 (82.53-95.30) at different time points”. The use of the word between requires a range of values (for instance, between 80 and 90). Otherwise you should use the word around or about. Moreover, I would not use the term “timepoints”, because time is not the independent variable of interest, I would rather use: “at different ECBF levels”.
(5) Page 7, line 204. Please correct “increase in both” instead of “increased of both”.
(6) Page 7, line 207. Much larger cannulas can also be inserted percutaneously. I would not state this as an advantage.
(7) Page 7, line 236 “swine models of healthy lungs”: Should probably be “swine models with healthy lungs”
Author Response
Reviewer 3
In this experimental study, De Nardo and colleagues evaluate the CO2 removal capacity of a novel device in a swine model. The principal interest of the study lies in the evaluation of this new device. It does not really add new generalizable information to the existing body of knowledge on extracorporeal CO2 removal.
MAJOR COMMENTS
(1) Comparisons with other ECCO2R membrane performances in the literature should be more thorough, detailed and explicit in the discussion. For instance, authors demonstrate a membrane lung VCO2 of 150 ml/min with a moderate blood flow of 600 mL/min. In a similar experiment, Duscio E and colleagues were able to demonstrate a similar VCO2 with a membrane of the same surface area with an even lower ECBF (400 ml/min). (Duscio E et al. Extracorporeal CO2 Removal: The Minimally Invasive Approach, Theory, and Practice, Crit Care Med. 2019 Jan;47(1):33-40.doi: 10.1097/CCM.0000000000003430)
We are grateful with the Reviewer for his revision of our paper. The paper of Duscio et al. has been included in our discussion
(2) A limitation of the study is that its design maximizes the membrane VCO2 by creating a very high inflow PaCO2. For a given ECBF, membrane surface area, and sweep gas flow, the VCO2 of a membrane lung will be highly dependent on the inflow PaCO2. (the higher the inflow PaCO2, the higher the VCO2). (Duscio et al CCM 2019) In their experiment, De Nardo and colleagues maintained inflow PaCO2 high (about 100 mm Hg) by the addition of CO2 to the inspired gas. This will result in an overestimation of VCO2 to be expected in a real life scenario, where clinicians would rarely aim to maintain a pH of 7.19 and a PaCO2 at 100 mmHg. This should be added to the limitations section of the discussion.
Thanks for this comment. This is an important message which we have been included in the limit of the study.
We are aware that the main indications of ECCO2R use are to potentially reduce of VILI in ARDS and avoid intubation in BPCO patients. However, there are many other patients which potentially could benefit of this device. For example, acute exacerbation of asthma, acute exacerbations in patients with end stage respiratory disease (e.g cystic fibrosis). These patients often reach very high CO2 (> 100 Hg) that could be managed with ECCO2R (maybe in the respiratory medicine wards), when medical therapy fails before moving to ECMO.
MINOR COMMENTS
(3) Page 5, line 156: “Even though not significant, […]”. What test was used? Please provide p-value in the text. The test used was the friedman test. P-value have been provided
(4) Page 5, line 158: “values remained between 85.95 (82.53-95.30) at different time points”. The use of the word between requires a range of values (for instance, between 80 and 90). Otherwise you should use the word around or about. Moreover, I would not use the term “timepoints”, because time is not the independent variable of interest, I would rather use: “at different ECBF levels”. These parts were fixed according to the reviewer suggestions
(5) Page 7, line 204. Please correct “increase in both” instead of “increased of both”. This has been corrected in the revised manuscript
(6) Page 7, line 207. Much larger cannulas can also be inserted percutaneously. I would not state this as an advantage. This sentence has been removed in the revised manuscript
(7) Page 7, line 236 “swine models of healthy lungs”: Should probably be “swine models with healthy lungs”
This has been corrected in the revised manuscript

Round 2
Reviewer 1 Report
I am happy to see that the paper has been enriched by the authors. I think that, despite the experimental setup is somehow not rigorous, given the extremely wide ranges of tolerated starting values of end tidal and PCO2 pre, the paper now has enough information to be suitable for publication.
I appreciate Table 1. I think that the authors should provide the values for each single animal so that the reader can follow the trajectory of each single subject. It is one of the few benefits of having such a small population.
Reviewer 3 Report
We are satisfied with this revision of the manuscript in terms of content.
We suggest the manuscript be reviewed for language structure and style.
For instance, page 3, paragraph 2: "a significant better reduction" should be a "significantly better reduction". "However, both studies convey on the point that the sweep gas flow can increase CO2 removal only when high blood flow rates are used.” We think perhaps "convey" is the appropriate word here. Do the authors mean "converge"?